# Application of the Theoretical Framework of Acceptability to assess a telephone-facilitated health coaching intervention for the prevention and management of type 2 diabetes

Linda Timm [1,2]*, Kristi Sidney Annerstedt[2], Jhon Álvarez Ahlgren[3], Pilvikki Absetz[4], Helle Mølsted Alvesson[2], Birger C. Forsberg[2], Meena Daivadanam [2,5,6]

1 Department of Neurobiology, Care Sciences and Society, Karolinska Institutet, Huddinge, Sweden, 2 Department of Global Public Health, Karolinska Institutet, Stockholm, Sweden, 3 Department of Learning, Informatics, Management & Ethics, Karolinska Institutet, Stockholm, Sweden, 4 Collaborative Care Systems Finland, Helsinki, Finland, 5 International Maternal and Child Health Division, Department of Women's and Children's Health, Uppsala University, Uppsala, Sweden, 6 Department of Food Studies, Nutrition and Dietetics, Uppsala University, Uppsala, Sweden

* Linda.timm@ki.se

**Data Availability Statement:** The data that support the findings of this study are available as a

## Abstract

### Background

Lifestyle interventions focusing on diet and physical activity for the prevention and management of type 2 diabetes have been found effective. Acceptance of the intervention is crucial. The Theoretical Framework of Acceptability (TFA) developed by Sekhon et al. (2017) describes the multiple facets of acceptance: Affective attitude, burden, perceived effectiveness, ethicality, intervention coherence, opportunity costs and self-efficacy. The aims of this study were to develop and assess the psychometric properties of a measurement scale for acceptance of a telephone-facilitated health coaching intervention, based on the TFA; and to determine the acceptability of the intervention among participants living with diabetes or having a high risk of diabetes in socioeconomically disadvantaged areas in Stockholm.

### Methods

This study was nested in the implementation trial SMART2D (Self-management approach and reciprocal learning for type 2 diabetes). The intervention consisted of nine telephone-facilitated health coaching sessions delivered individually over a 6-month period. The acceptability of the intervention was assessed using a questionnaire consisting of 19 Likert scale questions developed using Sekhon's TFA. Exploratory factor analysis (EFA) was performed.

### Results

Ratings from 49 participants (19 with type 2 diabetes and 30 at high risk of developing diabetes) in ages 38–65 were analyzed. The EFA on the acceptability scale revealed three factors

supplementary file: S5 Appendix SMART2D Acceptability raw data.

**Funding:** This study is part of the SMART2D project supported by the EU Horizon 2020 Health Coordination Activities (Grant Agreement No 643692), under call HCO-05-2014 (Global Alliance for Chronic Diseases: Prevention and treatment of type 2 diabetes) URL: https://ec.europa.eu/programmes/horizon2020/en/h2020-sectionsprojects; and partially funded by Region Stockholm's Public Healthcare Services Administration (Hälso- och sjukvårdsförvaltningen), URL: https://www.sll.se/omregionstockholm/Organisation/forvaltningar/halso-och-sjukvardsforvaltningen/. The funders had no role in study design, data collection and analysis, decision to publish, or preparation of the manuscript. The funders provided support for part of the salaries of authors [LT, KSA, JÁA, PA, HMA and MD], but did not have any additional role in the study design, data collection and analysis, decision to publish, or preparation of the manuscript. The specific roles of these authors are specified in the 'author contributions' section.

**Competing interests:** Collaborative Care Systems Finland (CCSF) is a social innovation enterprise classified as a small and medium enterprise (SME) partner in the SMART2D Consortium. Similar to other partners in the project, its work was funded by the European Commission Horizon 2020 grant. Apart from this project, CCSF provides services for fee in the form of consultation and training for health care organisations and professionals in implementing behaviour change science into practice in health promotion and disease prevention and self-management support. CCSF as a commercial body had no role in the study design, data collection and analysis, decision to publish, or preparation of the manuscript; it also does not alter our adherence to PLOS ONE policies on sharing data and materials.

with acceptable reliabilities: affective attitude (alpha 0.90), coherence and understanding (alpha 0.77), perceived burden (alpha 0.85), explaining 82% of the variance. Positive affect and coherence had high median scores and small variance. Median score for perceived burden was low, but with significant variance due to younger individuals and those at high risk reporting higher burden.

## Conclusions

The telephone-facilitated health coaching intervention was perceived as acceptable by the study population using a questionnaire based on Sekhon's TFA, with a wider variation in perceived burden seen among high risk and younger participants.

## Introduction

Lifestyle interventions focusing on diet, physical activity and(or) self-care behaviors for the prevention and management of type 2 diabetes (T2D) have been shown effective in improving cardio-metabolic outcomes [1, 2] in different population groups [2, 3]. Low participation rate and drop-outs are often a problem, suggesting issues related to accessibility and acceptance [4, 5]. This can be particularly challenging in groups with low socioeconomic status [6], where other social problems may require more attention [7]. Telephone coaching to support self-management is one strategy to overcome problems related to access [8–10].

Self-management of T2D leads to improvements in glycemic control [1]. Social support and education to manage the disease have been shown to improve self-management [11–13]. Improvements in health behavior, self-efficacy and health status among persons with chronic conditions (such as T2D, congestive cardiac failure, coronary artery disease, chronic obstructive pulmonary disease and hypertension) were reported in a systematic review on telephone-based interventions [8]. In this study, it was shown that telephone coaching was particularly beneficial to vulnerable populations, defined as individuals with low socioeconomic status, culturally and linguistically diverse backgrounds, low access to health services and often with worse control of their chronic conditions during baseline. Such interventions tend to have multiple components and are therefore complex, both in terms of their design and implementation.

Successful implementation of complex health interventions depend on many factors including acceptance of the intervention, and acceptability is seen as key when designing feasibility studies [14]. Interventions experienced as less intrusive have been reported as more acceptable [15], and studies suggest this also to be case with telephone coaching [9, 15]. However, there are examples of in-person coaching reported as more comfortable than coaching over telephone [9]. Comfort is, however only one aspect of acceptability. Acceptability has been defined as "a multi-faceted construct that reflects the extent to which people delivering or receiving a healthcare intervention consider it to be appropriate, based on anticipated or experienced cognitive and emotional responses to the intervention" [16]. While acceptability has long been recognized as an important criterion for user acceptance [17–19], it has been relatively underinvestigated [19], and there is no consensus on how to evaluate it.

Based on a review of systematic reviews, Sekhon et al. (2017) [16] developed the Theoretical Framework of Acceptability (TFA) with seven domains; affective attitude, burden, perceived effectiveness, ethicality, intervention coherence, opportunity costs and self-efficacy. The TFA has since been used to evaluate a telephone support intervention with automated calls to

improve self-management for persons with kidney disease [9]. The study used four of Sekhon's seven domains [16] to report both their quantitative and qualitative findings, and deemed telephone coaching to be an acceptable intervention method [9]. Although Sekhon's model on acceptability has been used to report the acceptability of interventions [9], the literature on quantitative assessments of acceptability using this model is scarce, and only a few quantitative tools have been developed based on the model. The aims of our study therefore were to: 1) develop and assess the psychometric properties of a measurement scale for acceptance of telephone-facilitated health coaching intervention, based on the TFA; and 2) determine the acceptability of the intervention among participants living with diabetes or having a high risk of developing diabetes in socioeconomically disadvantaged areas in Stockholm using the newly developed tool based on the TFA.

## Material and methods

This study was nested in SMART2D (Self-management approach and reciprocal learning for type 2 diabetes) [ISRCTN 11913581], a 5-year project (2015–2019) on implementation of contextualized self-management support in Sweden, South Africa and Uganda [20]. This study was conducted as part of the feasibility trial implemented in the Swedish arm of the SMART2D. Approval for the SMART2D trial protocol was given by the Regional Ethical Review Board in Stockholm effective 20th February 2017 (2016/2521-31/1). Written informed consent was obtained from each participant prior to enrollment in the SMART2D feasibility trial and permission was sought prior to the acceptability study. The Template for Intervention Description and Replication (TIDieR) checklist has been used for reporting (S1 Appendix).

### Setting

The study was implemented in two socioeconomically disadvantaged communities of Stockholm. These areas were characterized by low income levels and high unemployment rates. Compared to the overall Stockholm county, these areas have a high proportion of immigrants (Table 1).

**Participant recruitment.** 265 participants were recruited to participate in the SMART2D feasibility trial from two sources; 1) open community screening in public spaces at different time points and 2) out-patient lists from the primary healthcare centers. The participants had either a diagnosis of T2D within the last five years, or were identified as high risk of developing diabetes at the time of recruitment i.e., had a diagnosis of prediabetes or a score >13 on the Finnish Diabetes Risk Score (FINDRISC) [21, 22]. The feasibility trial used a cluster randomized design and 131 participants were included in the intervention arm. Of these, 72 participants (T2D: 29; high-risk: 43) received the telephone-facilitated health coaching intervention and the remaining were lost to follow-up.

**Swedish SMART2D intervention.** Development of the intervention as well as final intervention components have been described in detail elsewhere [4]. The intervention consisted of

**Table 1. Study setting characteristics.**

| Study area | Site 1 | Site 2 | Stockholm county |
|---|---|---|---|
| **Proportion of immigrants** (persons born outside Sweden or native born with both parents born outside Sweden) | 88.3% | 61.1% | 33.3% |
| **Unemployment rate** | 8% | 6.1% | 3% |
| **Income level SEK/year** | 204,600 | 245,600 | 374,400 |

SEK: Swedish kronor.

**Table 2. Overview of session structure in the telephone-facilitated health coaching intervention.**

| Session | Title | Content |
|---------|-------|---------|
| 1 | Introductory session | Getting to know the program. Why work with a care companion to make lifestyle changes? |
| 2 | Increase physical activity in daily life and reduce sedentary lifestyle | The importance of physical activity and how this can be increased in daily life |
| 3 | Healthy eating: Regular, balanced and healthy | The importance of regular, balanced and healthy meals |
| 4 | Physical activity through the life course | Discussion on how physical activity levels have changed over the years |
| 5 | Fruit and vegetables | The importance of eating fruit & vegetables every day |
| 6 | Increasing your daily physical activity | Discussion on current situation and potential possibilities for improvements |
| 7 | Sugar | How sugar consumption can be decreased in daily life |
| 8 | Finding a physical activity that suits you | Discussion of options/choices to physical activity |
| 9 | Healthy lifestyle—moving forward | How has it been to try to change to a healthier lifestyle and how can this be maintained? |

nine telephone-facilitated health coaching sessions (Table 2). The sessions were delivered individually by trained facilitators (SMART2D team members) on a weekly basis during the first five weeks and biweekly thereafter. The delivery period for the intervention was six months in total, from November 2018 to May 2019. During this period, the participants received structured support on lifestyle related habits over phone. An additional component of the intervention was to encourage social support through care companions (family members, friends or peers in the participant's close social network), with the aim of setting up goals and engaging in health-related activities together. Except for the introductory and concluding sessions, the remaining sessions focused alternately on diet and physical activity (Table 2). The facilitators followed a structured topic guide for each session based on the Motivational Behavioral Coaching (MBC) approach [23]. The participants were encouraged to set goals towards small changes by building on healthy habits and activities they were already familiar with. The focus was on positive affective processes [24, 25] i.e., working towards a specific goal with positive emotions which makes it more likely for the new health behavior to be maintained [24]. The sessions always started with a follow-up about the previous session and included a discussion on their goals and the next steps for further improvement. Every session ended with a question about the content of the session.

The sessions were delivered in Swedish, mostly using 'easy Swedish' with no use of medical or technical terms. When the participants' Swedish language skills were too limited to receive the intervention in Swedish, the intervention was delivered in the participants' native language by language skilled facilitators in Arabic, Somali or Spanish. Some participants preferred to have the intervention delivered in English, which was offered by all facilitators. The median duration per session was 19 minutes (range: 12–25 min) (Table 3).

In addition, two physical meetings per study site were arranged in the local community where the participants had the opportunity to meet each other, the facilitators, representatives from primary care, collaborating community organizations, experts or practitioners in diabetes, diet and physical activity, and other SMART2D team members. The meetings lasted approximately 3,5 hours each.

**Acceptability tool.** Questions were developed for each of the seven dimensions of Sekhon's acceptability framework [16] in relation to the SMART2D intervention. The

**Table 3. Duration per session.**

| Session | Median duration (50%) | Range (25%-75%) | Observations |
|---|---|---|---|
| All sessions | 19 | 12–25 | 308 |
| 1 | 11 | 9–12 | 45 |
| 2 | 19 | 15–23 | 47 |
| 3 | 22.5 | 18–27 | 44 |
| 4 | 19 | 16–25 | 33 |
| 5 | 20 | 13.5–25.5 | 32 |
| 6 | 16.5 | 12–23.5 | 28 |
| 7 | 19.5 | 15–23 | 26 |
| 8 | 19 | 13–27 | 24 |
| 9 | 26 | 21–35 | 29 |

instrument aimed to assess the acceptability of the intervention through a total of 19 questions (Fig 1).

The questionnaire used a 5 point Likert scale, where responses ranged from 5 (strongly agree) to 1 (strongly disagree). The content validity of the scale was evaluated by four researchers in the areas of public health, complex behavior change interventions, implementation science and medical anthropology and subsequently piloted with four participants.

## Data collection

At the end of the last telephone-facilitated health coaching session (session 9), the participants were asked for their consent to answer questions about the intervention later on during the same week. The acceptability survey was thereafter administered by a newly recruited research

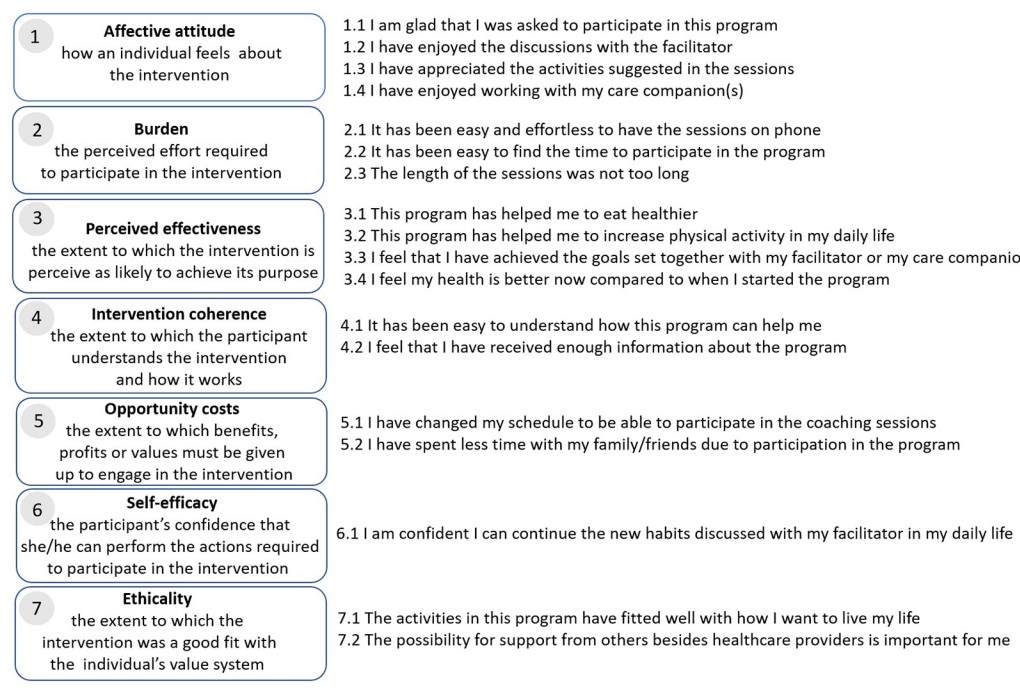

**Fig 1. Questions related to the seven domains in Sekhon's Theoretical Framework of Acceptability.**

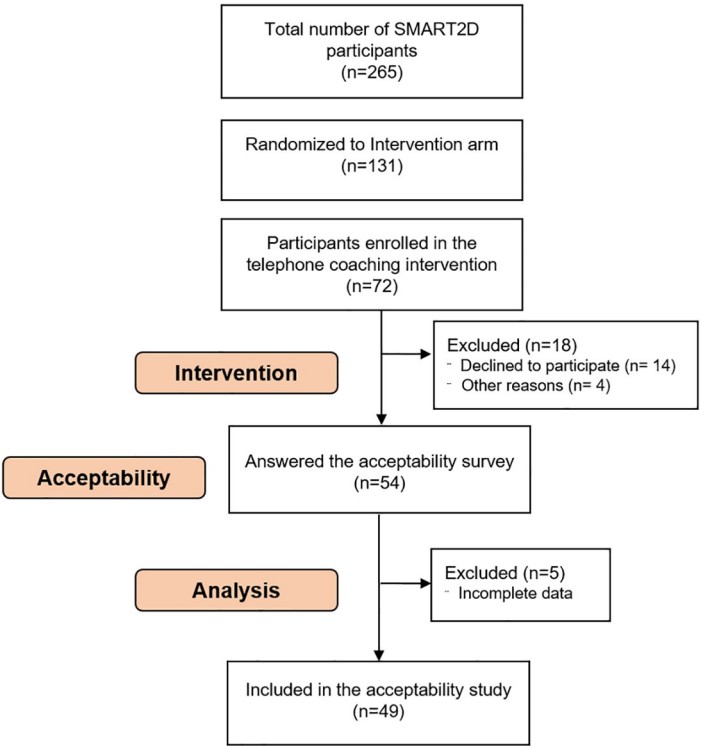

**Fig 2. Flow chart describing participant sampling for assessing acceptability of the SMART2D intervention.**

assistant with no prior knowledge about the project or contact with any of the participants. In addition participants who did not complete the intervention (< 3 sessions) were invited to fill in the acceptability survey. The survey took approximately ten minutes to complete. Out of the 72 participants who started the intervention, 54 participated in the acceptability survey. Five were excluded during the analysis due to incomplete responses. Data related to 49 participants, who had all completed the intervention, were included in the analyses for this study (response rate, 69%) (Fig 2).

## Data analysis

Descriptive statistics were computed to explain variable distributions. The scores of negatively worded statements were reversed in order to align the direction of the scale. Criterion validity was determined through Spearman's rank correlation coefficients, a non-parametric correlation test that is less susceptible to outliers, constructed between different items from the questionnaire. Exploratory factor analysis (EFA) was performed to assess the construct validity (discriminant and convergent validity) of the acceptability domains from the Sekhon framework [16]. Oblique (promax) rotations were selected to identify structure patterns and interpret the eigenvalues. The parallel analysis method was used to determine the number of factors to be retained. The Kaiser-Meyer-Olkin (KMO) measure of sampling adequacy was performed to confirm the appropriateness of data for the EFA [26] (KMO = 0.6230). Items with maximum loadings less than 0.40 were dropped. The internal consistency of the tool was assessed using Cronbach's alpha. Scores between 0.7 and 0.9 were deemed appropriate [27].

Likert summated scales were calculated for each participant within the identified construct from the EFA. Since the scales had a different number of items per construct, the sum-scales

were standardized to range from 0 to 100 [28]. Wilcoxon rank-sum test was conducted to detect differences between the sum-score for each domain and the socio-demographic characteristics. The statistical software package STATA version 15.1 was used to conduct the analysis.

## Results

Characteristics of the study population in comparison with those who did not conduct the acceptability survey are shown in Table 4. The majority of respondents were at high risk of developing diabetes (62%) and the age was in general lower in the high-risk group compared to participants with diabetes. In general, we had more female participants in the trial including the intervention arm and this pattern held true for participants and non-participants of the acceptability trial. The majority of participants were born outside Sweden (67%). The unemployment rate was higher in the high-risk group, which was also shown in the lower monthly income level in this group. No significant differences were found in employment status and income level between the participant groups. Therefore, no further analysis on the factor score distribution in relation to these variables were conducted. The participants in the acceptability

**Table 4. Participant characteristics.**

| | Diabetes n (%) | High-risk/ prediabetes or diabetes | Total from the Acceptability study | Participants who did not conduct the Acceptability survey | p-values |
|---|---|---|---|---|---|
| | **n = 19** | **n = 30** | **n = 49** | **n = 23** | |
| | 19 (38) | 30 (62) | | | |
| **Sex** | | | | | |
| Female | 10 (53) | 25 (83) | 35 (71) | 14 (61) | 0.370^ |
| Male | 9 (47) | 5 (17) | 14 (29) | 9 (39) | |
| **Age***  | 58 (48–65) | 44 (38–57) | 49 (40–60) | 53 (46–64) | 0.129† |
| Younger | 46 (40–49) | 39 (36–43) | 41(36–44) | 45 (41–46) | 0.231† |
| Older | 64 (58–69) | 59 (57–69) | 60 (57–69) | 60 (53–66) | 0.834† |
| **Household monthly Income (SEK)*** | 32,500 (22,000–44,800) | 20,000 (10,000–35,000)** | 29,500 (16,000–41,500)*** | 25,000 (13,000–45,000)‡ | 0.7334† |
| **Employment Status** | | | | | |
| Employed | 13 (68) | 16 (54) | 29 (59) | 11 (48) | 0.481^ |
| Unemployed/Unpaid work/ Supported by social services | 3 (16) | 10 (33) | 13 (27) | 6 (26) | |
| Retired | 3 (16) | 4 (13) | 7 (14) | 6 (26) | |
| **Number of intervention contacts*** | | | | | |
| 3 or more | 19 (100) | 28 (93) | 47 (96) | 2 (9) | 0.000^ |
| Less than 3 | 0 (0) | 2 (7) | 2 (4) | 21 (91) | |
| Median (IQR) | 9 (4–10) | 8 (4–10) | 8 (4–10) | 1 (0–1) | 0.000† |
| **Place of birth** | | | | | |
| Sweden | 10 (53) | 6 (20) | 16 (33) | 5 (21) | 0.342^ |
| Outside Sweden | 9 (47) | 24 (80) | 33 (67) | 18 (79) | |

*median,

^ Chi-square test,

† Kruskal-Wallis test,

**n = 21,

***n = 40,

‡ n = 19, SEK: Swedish kronor.

IQR: Interquartile range.

p-values compare the difference between the acceptability and excluded study sample.

trial and those who did not participate differed significantly only in terms of the number of intervention contacts.

## Tool reliability

In total 19 items were initially included in the EFA. Two items with maximum factor loadings less than 0.40 were excluded: 'It has been easy to find the time to participate in the program' and 'I have enjoyed working with my care companion(s)'. As shown in Table 5, after eliminating those items, three dimensions met extraction criteria and were retained: 1) Affective attitude and effectiveness (11 items on how the participants felt about the intervention and the extent to which the intervention was perceived as likely to achieve its purpose), 2) Coherence and understanding (4 items describing the extent to which the participants understood the intervention, how it addressed their condition and how it worked), and 3) Perceived burden (2 items on the perceived amount of effort that was required to participate in the intervention). The three factors accounted for 51%, 17% and 14% of the variance, respectively, giving a total of 82%. The Cronbach's alpha for the respective constructs were 0.90, 0.77 and 0.85 respectively indicating an appropriate level of internal consistency for each construct [29] (Table 5).

## Acceptability of the telephone-facilitated health coaching intervention

An overview of the factor score distribution is given in Table 6. Responses to the acceptability questionnaire (in %) as reported by the participants are tabulated in S4 Appendix. The analysis

**Table 5. Factor loadings from exploratory factor analysis and respective Cronbach's alpha scores for the final domains.**

|  | Affective attitude and effectiveness | Coherence and understanding | Perceived burden |
|---|---|---|---|
| This program has helped me to eat healthier | 0,91 |  |  |
| The possibility for support from others besides healthcare providers is important for me | 0,88 |  |  |
| I have enjoyed the discussions with the facilitator | 0,87 |  |  |
| This program has helped me to increase physical activity in my daily life | 0,68 |  |  |
| It has been easy to understand how this program can help me | 0,68 |  |  |
| I have appreciated the activities suggested in the sessions | 0,66 |  |  |
| I feel my health is better now compared to when I started the program | 0,65 |  |  |
| The activities in this program have fitted well with how I want to live my life | 0,55 |  |  |
| I am confident I can continue the new habits discussed with my facilitator in my daily life | 0,49 |  |  |
| I am glad that I was asked to participate in this program | 0,47 |  |  |
| I feel that I have achieved the goals set together with my facilitator or my care companion | 0,41 |  |  |
| I feel that I have received enough information about the program |  | 0,96 |  |
| The length of the sessions was not too long |  | 0,81 |  |
| I feel that I have received enough information about SMART2D |  | 0,60 |  |
| It has been easy and effortless to have the sessions on phone |  | 0,50 |  |
| I have spent less time with my family/friends due to participation in the program |  |  | 0,81 |
| I have changed my schedule to be able to participate in the coaching sessions |  |  | 0,76 |
| Eigenvalue | 5,8 | 3,7 | 2,0 |
| Variance explained | 47% | 30% | 16% |
| Cronbach's alpha | 0.90 | 0.77 | 0.85 |

Note: Factor loadings < 3 are omitted from the table. (R) indicates that the item is reversely scored.

**Table 6. Factor score distribution (median & interquartile range) for the final domains.**

| | Affective attitude and effectiveness | Coherence and understanding | Perceived burden |
|---|---|---|---|
| | Median (IQR) | Median (IQR) | Median (IQR) |
| Total | 91 (84–100) | 100 (81–100) | 0 (0–75) |
| n = 49 | | | |
| **Comparison between diagnostic groups** | | | |
| Diabetes | 87 (84–95) | 100 (88–100) | 0 (0–0) |
| n = 19 | | | |
| High risk | 95 (86–100) | 100 (81–100) | 38 (0–88) |
| n = 30 | | | |
| P-value | 0.1196 | 0.7078 | 0.0036 |
| **Comparison between age groups**[*] | | | |
| Younger | 97 (89–100) | 100 (81–100) | 38 (0–88) |
| n = 26 | | | |
| Older | 86 (80–93) | 100 (94–100) | 0 (0–38) |
| n = 23 | | | |
| P-value | 0.0066 | 0.5910 | 0.0515 |

IQR: Interquartile range;

[*] Age groups: younger: < median age and older: >/ = median age

showed high median scores for the standardized Likert summative scales, with a narrow inter-quartile range (IQR) for *affective attitude and effectiveness*, and *coherence and understanding* indicating a high acceptability of the intervention in terms of these two constructs. The opposite was seen in the third construct, *perceived burden*, where the median scores were low and with wide IQR, indicating an overall low perceived burden but with a wider variation in the responses. For the affective attitude and coherence and understanding, there were no significant differences found between the diagnostic groups (diabetes vs. high risk). The burden was perceived to be significantly higher among the participants in the high-risk group compared to those with T2D, and among younger participants compared to the older ones. Since high-risk participants were in general younger than the participants with diabetes (Table 4), the results were further tested for potential confounding (results not shown in tables). Both age and diagnostic groups remained, independent of each other, significant for perceived higher burden.

## Discussion

The tool based on Sekhon's model assessed the acceptability of the SMART2D intervention using three constructs: 1) Affective attitude and effectiveness; 2) Coherence and understanding; 3) Perceived burden. Acceptability of the SMART2D intervention was high for the first two constructs (affective attitude and coherence and understanding). Although the perceived burden remained relatively low among all participants, there were more variation with younger individuals and those at high-risk, showing a higher perceived burden compared to older individuals and those with T2D respectively.

### Affective attitude and effectiveness

The findings suggest a strong overall positive affect construct which includes affective elements related to the intervention process as well as to the outcomes [30]. This construct alone explained 47% of the variance in the acceptance measure (Table 5), which indicate that the intervention can be seen as acceptable from this standpoint. However, it also raises the question of whether positive affect, as induced by the contact with the facilitator and by the

outcomes of the coaching, is a sufficient measure for acceptance. Even though the telephone coaching was found acceptable, some participants would have preferred in-person coaching [9]. At the same time, the telephone coaching sessions provided by SMART2D were tailored to the individual. Also, a sense of relationship was established between facilitators and participants early on in the process as a high proportion of participants reported enjoying their discussions with the facilitator (S2 Appendix). Both of these aspects could be advantages for telephone-facilitated coaching compared to automated calls [9, 31]. The difference in age groups reported in Table 6 showed that scores for this construct was significantly higher among younger persons compared with the older participants.

## Coherence and understanding

The purpose of the health coaching program was perceived as clear and relevant to the participants which indicates that both the information about the project SMART2D and session content were sufficient for them to understand and internalize the objective of the intervention, making it possible to operationalize it in practice [32]. Operationalization of a theorized or conceptualized activity further brings into play a host of unknown variables and interactions, pertaining to the intervention itself, the intended population and the context in which the activity is implemented [4, 33–37]. Different starting points for lifestyle behaviors in terms of awareness and current practices [38, 39] and the different environmental cues and social support for these activities [11] could also potentially influence this domain. Duration and frequency of sessions have been found important in health coaching interventions [40]. A synthesis of reviews and meta-analyses considering health education related interventions mostly delivered face-to-face concluded that the recommended duration of sessions for patients with T2D should be more than 30 min per time per week [41]. One question within this domain addressed the participants' perception of the duration of the sessions, but did not clarify whether the duration of the sessions could have been considered as too short. We cannot therefore assess whether the participants would in fact have preferred longer sessions.

## Perceived burden

While the overall score for this construct was low, the intervention was perceived as more burdensome by individuals in the high-risk group (Table 6). Participants with T2D in our study were already familiar with lifestyle advice through their primary health care centers. As they were most likely already engaged at least in some activities for lifestyle modification, the intervention per se was not perceived as a burden to the same extent as those with high risk, for whom the activities may have been new. Moreover, persons at risk of diabetes in general are unlikely to have considered themselves to be ill [38, 39]. Therefore, all efforts towards lifestyle modification could have been perceived as something extra and requiring a greater effort [38, 39]. At the same time this group may be in greater need of support for lifestyle modification, since they do not have the same opportunity for support from primary care as individuals with T2D [42, 43]. Although the affective attitude and effectiveness was high among younger participants (Table 6), the intervention was perceived to be more burdensome among younger participants. Similar findings confirming a high burden for younger persons, were found in other studies showing higher participation numbers for older compared to younger participants in lifestyle interventions for diabetes prevention [44]. Older participants are also more likely to adhere to laboratory tests and have higher self-monitoring rates and adherence for taking medication [44, 45]. Younger participants have also shown to have poorer glucose control and persons with early-onset of T2D in younger ages have a higher prevalence of most clinical and behavioral risk factors, such as physical inactivity, tobacco smoking and obesity [46].

Both automated telephone coaching and in-person coaching have been found more effective for older participants [45]. Considerations to reduce the burden of participation to address the needs of younger participants [44] in terms of frequency [47], and tailoring of structure and content of interventions [48] are both indicated while designing interventions for lifestyle modification.

## Strengths and limitations

This study was nested in a feasibility implementation trial, making the results valid under real-life conditions for the target population and study setting. This has positive implications for further development and testing of this intervention among socioeconomically disadvantaged populations and suburbs in Sweden.

Although our empirical test on the 19 questions did not confirm the seven theoretically distinct constructs, it is most likely that all of them are represented in the three constructs as the dimensions are interlinked in practice and overlapping in peoples thinking and experience. There could potentially be more dimensions that we missed due to limited number of our participants. The study would benefit to be repeated with more participants to see if the results are replicated.

In terms of quality, the 5 point Likert scale format used in this study is preferred to 7 or 11 point scales [49]. However, one risk in agree-disagree scales is the acquiescence response bias, where it is more common that participants agree than disagree with the statement offered regardless of its content [50, 51]. That could have been the case in our study as well. Similarly, social desirability bias could also be relevant, with the acceptability survey being administered by the same project, of which they were a part. However, precautions were taken to minimize this bias through the collection of acceptability data by a research assistant, unfamiliar with the intervention and to the participants.

The participants of the acceptability survey had in general a higher number of intervention contacts compared to the non-participants (Table 4). This could indicate that the group who reported the acceptability was in general more positive to the intervention than those who discontinued the intervention for any reason. Qualitative interviews with participants who discontinued the intervention would have been valuable to understand how they viewed the intervention and its acceptability.

## Conclusions

We found that the telephone-facilitated health coaching intervention was perceived as acceptable by the study population in socioeconomically disadvantaged suburbs of Stockholm, using a tool based on Sekhon's theoretical framework of acceptability. Though acceptable, participation in a lifestyle intervention was found to be more burdensome for persons at high risk of developing diabetes and for younger participants. The individually tailored telephone coaching seemed therefore to be a suitable approach especially for people already living with type 2 diabetes in this population. This is also one of the initial attempts at a survey-based tool to assess acceptability for this type of behavior change interventions, and hence a work in progress, which needs to be refined further.

## Supporting information

**S1 Appendix. TIDieR-checklist.**
(DOCX)

**S2 Appendix. SMART2D intervention guide_points of contact and SOPs SWEDISH.**
(PDF)

**S3 Appendix. SMART2D intervention guide _points of contact and SOPs ENGLISH.**
(PDF)

**S4 Appendix. Reported responses.**
(DOCX)

**S5 Appendix. SMART2D acceptability raw data.**
(XLSX)

## Acknowledgments

We would like to acknowledge the contribution of Jeroen De Man throughout the study process and for reviewing the results of the analysis and Nicola Orsini for his guidance on the statistical analysis.

## Author Contributions

**Conceptualization:** Linda Timm, Helle Mølsted Alvesson, Meena Daivadanam.

**Data curation:** Linda Timm, Kristi Sidney Annerstedt, Jhon Álvarez Ahlgren.

**Formal analysis:** Linda Timm, Kristi Sidney Annerstedt, Jhon Álvarez Ahlgren, Pilvikki Absetz, Meena Daivadanam.

**Funding acquisition:** Meena Daivadanam.

**Investigation:** Linda Timm, Kristi Sidney Annerstedt, Jhon Álvarez Ahlgren, Pilvikki Absetz, Helle Mølsted Alvesson, Birger C. Forsberg, Meena Daivadanam.

**Methodology:** Pilvikki Absetz, Meena Daivadanam.

**Project administration:** Linda Timm, Meena Daivadanam.

**Resources:** Helle Mølsted Alvesson, Meena Daivadanam.

**Supervision:** Helle Mølsted Alvesson, Birger C. Forsberg, Meena Daivadanam.

**Validation:** Linda Timm, Kristi Sidney Annerstedt, Jhon Álvarez Ahlgren, Pilvikki Absetz, Helle Mølsted Alvesson, Meena Daivadanam.

**Visualization:** Linda Timm.

**Writing – original draft:** Linda Timm.

**Writing – review & editing:** Linda Timm, Kristi Sidney Annerstedt, Jhon Álvarez Ahlgren, Pilvikki Absetz, Helle Mølsted Alvesson, Birger C. Forsberg, Meena Daivadanam.

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
