## [Decision Letter · Decision Letter 0]

3 Dec 2020

PONE-D-20-32022

Application of the Theoretical Framework of Acceptability to assess a telephone-facilitated health coaching intervention for the prevention and management of type 2 diabetes

PLOS ONE

Dear Dr. Timm,

Thank you for submitting your manuscript to PLOS ONE. After careful consideration, we feel that it has merit but does not fully meet PLOS ONE’s publication criteria as it currently stands. Therefore, we invite you to submit a revised version of the manuscript that addresses the points raised during the review process.

We look forward to receiving your revised manuscript.

Kind regards,

Cindy Gray, Ph.D.

Academic Editor

PLOS ONE

Journal Requirements:

2.) Please include additional information regarding the coaching material used in the study and ensure that you have provided sufficient details that others could replicate the analyses. If you developed material for the telephone-facilitated health coaching session as part of this study and it is not under a copyright more restrictive than CC-BY, please include a copy, in both the original language and English, as Supporting Information, or include a citation if it has been published previously.

3.) We note that you have indicated that data from this study are available upon request. PLOS only allows data to be available upon request if there are legal or ethical restrictions on sharing data publicly. For information on unacceptable data access restrictions, please see http://journals.plos.org/plosone/s/data-availability#loc-unacceptable-data-access-restrictions.

4.) Thank you for stating the following in the Financial Disclosure section:

'This study is part of the SMART2D project supported by the EU Horizon 2020 Health Coordination Activities (Grant Agreement No 643692), MD, under call HCO-05-2014 (Global Alliance for Chronic Diseases: Prevention and treatment of type 2 diabetes) URL: https://ec.europa.eu/programmes/horizon2020/en/h2020-sections-projects; and partially funded by Region Stockholm’s Public Healthcare Services Administration (Hälso- och sjukvårdsförvaltningen), URL: https://www.sll.se/om-regionstockholm/Organisation/forvaltningar/halso-och-sjukvardsforvaltningen/. The funders had no role in study design, data collection and analysis, decision to publish, or preparation of the manuscript.'

We note that one or more of the authors are employed by a commercial company: Collaborative Care Systems Finland

5.) Please include captions for your Supporting Information files at the end of your manuscript, and update any in-text citations to match accordingly. Please see our Supporting Information guidelines for more information: http://journals.plos.org/plosone/s/supporting-information

Additional Editor Comments:

Abstract: it would be good to have participant numbers in the method, and a brief description of who the participants were: for example, age, gender and SES in the results. The conclusion could flow better.

Introduction: it would be good to reference the new scale in the second aim.

Method: How long is each intervention session? On lines 157 to 159, it is a bit confusing with regards to which language the sessions were presented in. More detail is required about the face to face meetings (for example, how often/how long?). In Table 2, Session 4, is the change over the years in relation to individual or population physical activity? The data analysis only appears to describe factor analysis and not application of the tool to assess acceptability (or if it does do the latter then could this be stated more explicitly?)

Results: it would be good to have employment status and income level for participants presented. And, if appropriate, do some analysis of the factor score distribution in relation to these.

Discussion: line 289, I am not sure of the point of discussing smiley faces – could its relevance be made clearer? It would have been useful to have some more detail of studies 44 and 45. Given the low participant number, should the study be repeated with more participants to see if the results are replicated?

Reviewers' comments:

Reviewer's Responses to Questions

**Comments to the Author**

1. Is the manuscript technically sound, and do the data support the conclusions?

Reviewer #1: Partly

2. Has the statistical analysis been performed appropriately and rigorously? 

Reviewer #1: No

3. Have the authors made all data underlying the findings in their manuscript fully available?

Reviewer #1: Yes

4. Is the manuscript presented in an intelligible fashion and written in standard English?

Reviewer #1: Yes

5. Review Comments to the Author

Reviewer #1: The data analysis section needs more clarity.

Line 199: Please specify the type of correlation used to estimate the correlation matrix and justify the choice.

Lines 203-204: The KMO test was used but the criteria used to determine appropriateness has not been specified.

Line 208: It is not clear what m and k stand for. This has not been defined in the analysis section.

Lines 208-209: Please clarify what differences were being tested using the Wilcoxon rank sum test and its role in evaluating the psychometric properties of the scale.

Please justify why were unidimensionality and validity (content, criterion, construct, convergent and divergent) not evaluated.

6. PLOS authors have the option to publish the peer review history of their article (what does this mean?). If published, this will include your full peer review and any attached files.

Reviewer #1: No

---

## [Author Response · Author response to Decision Letter 0]

15 Jan 2021

Dear Editors!

Our responses are provided in a table format uploaded as a separate document.

Please contact me if you want me to enter our responses in this text box (needs to be edited to fit in).

Thank you,

Linda Timm

linda.timm@ki.se

---

## [Decision Letter · Decision Letter 1]

10 Sep 2021

PONE-D-20-32022R1Application of the Theoretical Framework of Acceptability to assess a telephone-facilitated health coaching intervention for the prevention and management of type 2 diabetesPLOS ONE

Dear Dr. Timm,

Thank you for submitting your manuscript to PLOS ONE. After careful consideration, we feel that it has merit but does not fully meet PLOS ONE’s publication criteria as it currently stands. Therefore, we invite you to submit a revised version of the manuscript that addresses the points raised during the review process. To ensure completeness of methodological reporting, please provide the additional details requested by Reviewer 2. Please submit your revised manuscript by Oct 24 2021 11:59PM. If you will need more time than this to complete your revisions, please reply to this message or contact the journal office at plosone@plos.org. Please include the following items when submitting your revised manuscript:A rebuttal letter that responds to each point raised by the academic editor and reviewer(s). You should upload this letter as a separate file labeled 'Response to Reviewers'.A marked-up copy of your manuscript that highlights changes made to the original version. You should upload this as a separate file labeled 'Revised Manuscript with Track Changes'.An unmarked version of your revised paper without tracked changes. You should upload this as a separate file labeled 'Manuscript'.If applicable, we recommend that you deposit your laboratory protocols in protocols.io to enhance the reproducibility of your results. Protocols.io assigns your protocol its own identifier (DOI) so that it can be cited independently in the future. For instructions see: https://journals.plos.org/plosone/s/submission-guidelines#loc-laboratory-protocols. Additionally, PLOS ONE offers an option for publishing peer-reviewed Lab Protocol articles, which describe protocols hosted on protocols.io. Read more information on sharing protocols at https://plos.org/protocols?utm_medium=editorial-email&utm_source=authorletters&utm_campaign=protocols.

We look forward to receiving your revised manuscript.

Kind regards,

Jamie Males

Staff Editor

PLOS ONE

Journal Requirements:

Reviewers' comments:

Reviewer's Responses to Questions

**Comments to the Author**

1. If the authors have adequately addressed your comments raised in a previous round of review and you feel that this manuscript is now acceptable for publication, you may indicate that here to bypass the “Comments to the Author” section, enter your conflict of interest statement in the “Confidential to Editor” section, and submit your "Accept" recommendation.

Reviewer #2: (No Response)

2. Is the manuscript technically sound, and do the data support the conclusions?

Reviewer #2: Yes

3. Has the statistical analysis been performed appropriately and rigorously? 

Reviewer #2: Yes

4. Have the authors made all data underlying the findings in their manuscript fully available?

Reviewer #2: Yes

5. Is the manuscript presented in an intelligible fashion and written in standard English?

Reviewer #2: Yes

6. Review Comments to the Author

Reviewer #2: The aims of the cluster randomized design research study were to develop and assess the psychometric properties of measurement scale for acceptance of telephone-facilitated health coaching intervention based on Theoretical Framework Acceptability (TFA), and to determine the acceptability of the intervention among participants living with diabetes or having a high risk of diabetes in socioeconomically disadvantaged areas of Stockholm. The Exploratory Factor Analysis on the acceptability scale revealed three factors with acceptable reliabilities: affective attitude (alpha 0.90), coherence and understanding (alpha 0.77), perceived burden (alpha 0.85), explaining 82% of the variance.

Minor revision:

Cite the statistical software used for the analysis.

7. PLOS authors have the option to publish the peer review history of their article (what does this mean?). If published, this will include your full peer review and any attached files.

Reviewer #2: No

---

## [Author Response · Author response to Decision Letter 1]

18 Oct 2021

Dear Reviewer,

We are thankful for your suggestions on revisions to improve our manuscript.

Details on the revisions can be found in the attached file "Respons to Reviewers".

Thank you!

Kind regards,

Linda

---

## [Decision Letter · Decision Letter 2]

29 Jun 2022

PONE-D-20-32022R2Application of the Theoretical Framework of Acceptability to assess a telephone-facilitated health coaching intervention for the prevention and management of type 2 diabetesPLOS ONE

Dear Dr. Timm,

Thank you for submitting your manuscript to PLOS ONE. After careful consideration, we feel that it has merit but does not fully meet PLOS ONE’s publication criteria as it currently stands. Therefore, we invite you to submit a revised version of the manuscript that addresses the points raised during the review process.

We look forward to receiving your revised manuscript.

Kind regards,

Miquel Vall-llosera Camps

Senior Editor

PLOS ONE

Additional Editor Comments:

Apologies for the delay in providing this review and for the new reviews at this stage, but due to the clinical implications of this study it was considered necessary to invite additional reviewers to assess the manuscript. Reviewer 3 comments should be straightforward to address. Reviewer 5 comments about the introduction and discussion are optional (as these are not required to meet our publication criteria), but the comments about the methods and results are necessary and should be straightforward too.

Reviewers' comments:

Reviewer's Responses to Questions

**Comments to the Author**

1. If the authors have adequately addressed your comments raised in a previous round of review and you feel that this manuscript is now acceptable for publication, you may indicate that here to bypass the “Comments to the Author” section, enter your conflict of interest statement in the “Confidential to Editor” section, and submit your "Accept" recommendation.

Reviewer #2: All comments have been addressed

Reviewer #3: (No Response)

Reviewer #4: All comments have been addressed

Reviewer #5: (No Response)

2. Is the manuscript technically sound, and do the data support the conclusions?

Reviewer #2: (No Response)

Reviewer #3: Yes

Reviewer #4: Yes

Reviewer #5: Yes

3. Has the statistical analysis been performed appropriately and rigorously? 

Reviewer #2: (No Response)

Reviewer #3: Yes

Reviewer #4: I Don't Know

Reviewer #5: Yes

4. Have the authors made all data underlying the findings in their manuscript fully available?

Reviewer #2: (No Response)

Reviewer #3: Yes

Reviewer #4: Yes

Reviewer #5: Yes

5. Is the manuscript presented in an intelligible fashion and written in standard English?

Reviewer #2: (No Response)

Reviewer #3: Yes

Reviewer #4: Yes

Reviewer #5: Yes

6. Review Comments to the Author

Reviewer #2: (No Response)

Reviewer #3: This is an interesting paper on the psychometric property of TFA scale. Although the sample size is relatively small, the analytical approach to relevant theoretical constructs offers some evidence of the validity and reliability of the measurement indicators. The paper has adequately addressed the standard procedure used in the evaluation of psychometric property of measurement items.

Two suggestive amendments are as follows:

1. Exploratory Factor Analysis: Are you assuming that the five theoretical constructs are independent? Are the five constructs correlated with each other?

2. Limitation of EFA: Originally, there were seven constructs developed for TFA. However, only five constructs appear to show the relevance to TFA. The EFA procedure is sensitive to the variations in the samples selected. In other words, how can investigators handle the potential biases introduced by the sample?

Reviewer #4: Thank you for the opportunity to review. I would be happy to state that this publication may be accepted as is.

Reviewer #5: Thank you for the opportunity to review this manuscript on the acceptability of a telephone-facilitated health coaching intervention. It appears that this is a resubmission; however, I did not review the first version. Therefore, my comments are new to the authors. Overall, the manuscript’s methods and results are well described. The greatest attention in the revisions should be paid to the introduction framing the study and the discussion of its findings. I have summarized needed revisions by sections below.

Introduction:

The introduction needs some restructuring. An outline that would make more sense is: 1. Discuss diabetes and the self-management it requires. 2. Discuss self-management benefits for T2D. 3. Discuss challenges to doing self-management activities – these include social determinants of health, treatment burden, and patient capacity. 4. Discuss health coaching as an intervention to potentially overcome these challenges. 5. Discuss potential challenges to implementation of telephone health coaching, including acceptability. 6. Discuss acceptability specifically. 8. Discuss TFA as a way to assess acceptability and why it is useful. 7. Lay out the aims.

Line 71. First sentence needs citations.

Methods:

Please note the training of the trained facilitators specifically. Table 2, session content notes “peer” in the description. Did the study use peer coaches? If peers, it need to specifically state throughout that the intervention of interest is peer coaching. Peer coaching is not currently covered within the standardized definition of Health and Wellness Coaching, which is currently driving practice changes, including reimbursement, in the US Setting. (Wolever 2013 in Global advances in Health and Medicine; Wolever 2016 in BMC Health Services Research).

Please describe the amount of missingness led to surveys being excluded from the analysis (n=6).

Results:

I don’t think that “excluded from the acceptability study” is the right terminology. If I am reading methods correctly, only 6 were excluded due to missingness. The others chose not to participate in the acceptability study portion. Therefore, I think something like “intervention completed; declined acceptability study” is more appropriate. The 6 that didn’t fully complete the data collection of the acceptability study could still be labelled as excluded.

Discussion:

In the first paragraph, your summary of findings, please add a statement that relates to your first aim. Currently, the summary primarily focuses on the second aim.

Similarly, the discussion pays very little attention to the scale that was developed. While understandable that the scale served a purpose to evaluate the intervention, significant attention was devoted to it in the methods and results sections. It would be helpful to highlight future uses of the scale – is it only possible to use it with this specific intervention or could it be applicable (as is or tailored) to other coaching or self-management interventions as well?

You note in results that the intervention was more burdensome to younger individuals. This is in alignment with treatment burden in general. Across populations, younger patients report higher levels of treatment burden. See work by Tran VT and Eton DT. This should be noted in the section on burden.

It would be helpful to add a section that specifically addresses implications of the study for practice and for research.

7. PLOS authors have the option to publish the peer review history of their article (what does this mean?). If published, this will include your full peer review and any attached files.

Reviewer #2: No

Reviewer #3: **Yes: **Thomas T.H. Wan

Reviewer #4: No

Reviewer #5: **Yes: **Kasey R. Boehmer

---

## [Author Response · Author response to Decision Letter 2]

27 Aug 2022

Response to Reviewers

Thank you for your valuable comments. Please see our response and revisions below.

Reviewer #3: 

Two suggestive amendments are as follows:

1. Exploratory Factor Analysis: Are you assuming that the five theoretical constructs are independent? Are the five constructs correlated with each other?

Response to suggestive amendment 1: 

The items in the different theoretical constructs are correlated with each other as reported in the correlation matrix of the data. The original constructs are all part of the Theoretical Framework of Acceptability as discussed in the text. Further, the EFA has retained the most relevant items to our acceptability dimensions representing what we believe are the participants’ thinking and experience of the intervention.

2. Limitation of EFA: Originally, there were seven constructs developed for TFA. However, only five constructs appear to show the relevance to TFA. The EFA procedure is sensitive to the variations in the samples selected. In other words, how can investigators handle the potential biases introduced by the sample?

Response to suggestive amendment 2: 

It is possible that with a larger sample we would have been able to keep a larger number of items and therefore a more detailed representation of the original seven constructs of the TFA. However, the sampling adequacy test showed that the data used in our analyses was adequate to perform EFA, the items included in this study and the obtained constructs were relevant and showed validity to our acceptability study. 

Reviewer #5:

Methods: Please note the training of the trained facilitators specifically. Table 2, session content notes “peer” in the description. Did the study use peer coaches? If peers, it need to specifically state throughout that the intervention of interest is peer coaching. Peer coaching is not currently covered within the standardized definition of Health and Wellness Coaching, which is currently driving practice changes, including reimbursement, in the US Setting. (Wolever 2013 in Global advances in Health and Medicine; Wolever 2016 in BMC Health Services Research).

Response: Thank you for this valuable comment. It was not correct to use “peer” in Table 2. We have clarified this by replacing the word peer to the more adequate word care companion, which is also used in the descriptive text.

Please describe the amount of missingness led to surveys being excluded from the analysis (n=6). 

Response: Revision on page 9. To clarify the amount of missingness, the number is corrected to 5 in the text, now corresponding to the numbers in the figure.

Results: I don’t think that “excluded from the acceptability study” is the right terminology. If I am reading methods correctly, only 6 were excluded due to missingness. The others chose not to participate in the acceptability study portion. Therefore, I think something like “intervention completed; declined acceptability study” is more appropriate. The 6 that didn’t fully complete the data collection of the acceptability study could still be labelled as excluded.

Response: Thank you for this comment, the terminology is revised both in the text and in the table for clarification (pages 10-11). Please note the comment above on the revised number.

We have considered the comments about introduction and discussion from Reviewer 5. These revisions are according to editors optional and we have chosen to leave these parts as they have been thoroughly revised earlier according to previous reviewers comments. 

Line 71: Citation marks have been added to the first sentence.

Line 369: An ‘i’ was missing after the word interlinked and this is corrected.

Table 4 on page 1. ‘of’ has been changed to ‘or’.

---

## [Decision Letter · Decision Letter 3]

20 Sep 2022

Application of the Theoretical Framework of Acceptability to assess a telephone-facilitated health coaching intervention for the prevention and management of type 2 diabetes

PONE-D-20-32022R3

Dear Dr. Timm,

We’re pleased to inform you that your manuscript has been judged scientifically suitable for publication and will be formally accepted for publication once it meets all outstanding technical requirements.

Kind regards,

Kingston Rajiah

Academic Editor

PLOS ONE

Additional Editor Comments (optional):

Reviewers' comments:

Reviewer's Responses to Questions

**Comments to the Author**

1. If the authors have adequately addressed your comments raised in a previous round of review and you feel that this manuscript is now acceptable for publication, you may indicate that here to bypass the “Comments to the Author” section, enter your conflict of interest statement in the “Confidential to Editor” section, and submit your "Accept" recommendation.

Reviewer #2: All comments have been addressed

Reviewer #3: All comments have been addressed

2. Is the manuscript technically sound, and do the data support the conclusions?

Reviewer #2: (No Response)

Reviewer #3: Yes

3. Has the statistical analysis been performed appropriately and rigorously? 

Reviewer #2: (No Response)

Reviewer #3: Yes

4. Have the authors made all data underlying the findings in their manuscript fully available?

Reviewer #2: (No Response)

Reviewer #3: Yes

5. Is the manuscript presented in an intelligible fashion and written in standard English?

Reviewer #2: (No Response)

Reviewer #3: Yes

6. Review Comments to the Author

Reviewer #2: (No Response)

Reviewer #3: The revised submission has adequately addressed the issues. Hence, the readability of the paper has been enhanced.

7. PLOS authors have the option to publish the peer review history of their article (what does this mean?). If published, this will include your full peer review and any attached files.

Reviewer #2: No

Reviewer #3: **Yes: **Thomas T.H. Wan

---

## [Editor Report · Acceptance letter]

26 Sep 2022

PONE-D-20-32022R3 

Application of the Theoretical Framework of Acceptability to assess a telephone-facilitated health coaching intervention for the prevention and management of type 2 diabetes 

Dear Dr. Timm:

I'm pleased to inform you that your manuscript has been deemed suitable for publication in PLOS ONE. Congratulations! Your manuscript is now with our production department. 

Kind regards, 

on behalf of

Associate Professor Kingston Rajiah 

Academic Editor

PLOS ONE